# Exploring the Involvement of Personal and Emotional Factors and Social Media Body Image-Related Behaviours on Eating Disorder Symptoms and Body Image Concerns in Women and Men

**DOI:** 10.3390/healthcare13161997

**Published:** 2025-08-14

**Authors:** Celia López-Muñóz, Roberto García-Sánchez, Melany León-Méndez, Rosario J. Marrero

**Affiliations:** 1Department of Clinical Psychology, Psychobiology, and Methodology, Universidad de La Laguna, 38200 La Laguna, Spain; celialopez.psicologia@gmail.com (C.L.-M.); mleonmen@ull.edu.es (M.L.-M.); 2Facultad de Ciencia Biomédicas y de la Salud. Sección Psicología, Universidad Europea de Canarias, 38300 La Orotava, Spain; extrgarcias@ull.edu.es; 3Institute of Neuroscience, Universidad de La Laguna, 38200 La Laguna, Spain

**Keywords:** body image, eating disorders, social media, perfectionism, emotion dysregulation, psychological functioning

## Abstract

**Background**: Eating disorder (ED) symptoms and body image (BI) concerns involve serious risks to the physical and mental health of women and men. Social networking sites have amplified the promotion of idealised body images, contributing to this issue. **Objectives**: This study examines the link between personal and emotional factors, social media body image-related behaviours, BI concerns, and ED symptoms, as well as the differential role of these factors according to gender. **Methods**: A cross-sectional design was applied. The participants were 201 Spanish adults (mean age = 28.26; 76.6% women) who completed self-reported measures on BI (MBSRQ), ED symptoms (EAT-26), perfectionism (EDI-2), anxiety, depression, stress (DASS-21), emotional dysregulation (DERS), life satisfaction (SWLS), and social media BI-related behaviours. **Results**: Significant relationships between ED symptoms and BI concerns with personal and emotional factors and social media body image-related behaviours have been found. Women have scored higher in ED symptoms and social media BI-related behaviours than men. Multiple regression analyses showed that the difficulty in accepting emotions and not posting images due to BI dissatisfaction were risk factors for ED symptoms in both men and women. Furthermore, the adverse effect of perfectionism and low life satisfaction on women’s ED symptoms were demonstrated, whereas in men, goal-directed behaviours were associated with ED risk. **Conclusions**: These results suggest that prevention programmes focusing on emotional regulation and healthier social media use can be effective for ED symptoms and BI concerns.

## 1. Introduction

Eating disorders (EDs) are characterised as alterations or dysregulation of food consumption or intake, leading to a significant decline in physical health and psychological and social functioning [1]. There are different extreme manifestations along the continuum of EDs, such as bulimia or anorexia nervosa, as well as subclinical forms that are important to identify for preventive reasons. The global prevalence of EDs has doubled in recent decades, increasing from 3.5% to 7.8% [2]. Alternatively, body image (BI) is traditionally defined as a person’s perceptions, thoughts, and feelings about their body, shaped by sociological, physiological, and psychoanalytic perspectives [3]. BI encompasses attitudes and evaluations towards one’s body, which may not necessarily correspond to actual physical appearance [4]. BI has been consistently associated with ED symptomatology. Excessive concerns about BI can lead to attempts to alter one’s appearance, often through disordered eating behaviours [5]. Identifying factors associated with BI concerns is therefore essential to preventing unhealthy behaviours and more severe eating problems.

The growing use of social media has been shown to affect people’s behaviour, with negative effects on BI and EDs. Social networking sites amplify the promotion of idealised body image related to fashion, beauty, diets, and fitness, which exacerbates BI concerns and EDs [6,7,8]. Research indicates that the appearance-focused use of social media is linked to greater internalisation of thinness as an ideal of beauty, restrictive eating behaviours, social comparison, and an increased need for external validation [9]. Seeking information about weight management through eating and exercise habits is a prevalent activity on social networking sites [10]. This activity has been shown to have a significant impact on BI, EDs, and overall well-being [11]. Considering that social media is increasingly used for various purposes—such as searching for or exchanging information, entertainment, or social interaction—analysing the effects of social media on BI concerns and EDs symptoms is particularly relevant. BI concerns, combined with the pressure exerted on BI on social media, emerge as a pivotal risk factor in the onset and maintenance of EDs and related behaviours, particularly among women and adolescents [12].

Several personal, emotional, and social factors have been identified as increasing the vulnerability to BI concerns and ED symptoms. In this sense, perfectionism, in particular, is a personality trait that has been shown to be a risk factor for ED [13,14] and BI dissatisfaction [15]. It involves setting excessively high standards, worrying about making mistakes, and fearing negative evaluation [16]. A recent meta-analysis found that both perfectionistic striving and preoccupation were associated with EDs, and that these traits interfere with the prevention and treatment efforts [17]. Additionally, no differences in perfectionism have been observed across different EDs, suggesting that it may be a transdiagnostic factor, with a primary focus on concerns over mistakes [18,19]. Therefore, analysing the role of perfectionism in BI concerns and ED symptoms is particularly relevant for promoting more realistic standards related to physical appearance and eating, which can contribute to the prevention of these problematic behaviours.

An additional factor strongly associated with EDs is the emotional component. Emotion regulation refers to the processes through which individuals attempt to influence what emotions they experience, as well as when, how, and where they are experienced [20]. In contrast, affective dysregulation denotes a deficit in the activation and/or effectiveness of emotional regulation strategies resulting in maladaptive outcomes [21]. Difficulties in emotional regulation have been consistently linked to both aetiology and persistence of EDs [22,23]. Moreover, emotional dysregulation has been shown to mediate the relationship between perfectionism and preoccupation with shape, weight, and dietary restraint [24].

Other emotional states, such as anxiety, depression, and stress, have also been identified as significant contributors to EDs [25,26]. Stress is a well-established factor in ED development, as it often triggers negative emotions and disrupts normal eating patterns [27]. Emotional disturbances may manifest as either increased or decreased in caloric intake, and a significant correlation has been observed between emotional stress (e.g., anxiety, depression, stress) and unhealthy eating patterns [28]. Stress, in particular, is associated with the consumption of sugar-rich foods, often used as a coping mechanism to manage negative emotions [29]. Among female adolescents and young adults, depression and anxiety not only exacerbate ED-related impairment but also significantly moderate the relationship between ED symptoms and overall functional impairment [30].

Regarding protective factors, life satisfaction has been shown to be associated with fewer ED symptoms and BI concerns. In general, an adequate and balanced diet leads to greater life satisfaction compared to an altered intake—whether excessive or deficient, characteristic of eating disorders. A positive correlation has been identified between balanced nutritional intake and life satisfaction, indicating that individuals who maintain a healthy diet and feel satisfied with their eating habits generally report higher life satisfaction [31]. Similarly, young people who engage in healthy behaviours, such as physical exercise, tend to have more positive perceptions of their BI, which in turn leads to increased life satisfaction [32]. These results are in line with other studies that have demonstrated a positive correlation between healthy BI and psychological well-being [6]. Conversely, EDs are strongly linked to significant declines in quality of life, reflecting the profound negative impact these disorders can have on overall well-being [33].

Research suggests differences between women and men in EDs and BI concerns. In general, women report greater BI dissatisfaction and hold a more distorted perception of the ideal of beauty compared to men [34]. EDs also remain more prevalent among women [35], although recent years have seen a rise in the incidence of eating problems and diet-related concerns among [36]. Men are more likely to rely on physical exercise as a weight control method rather than adopting special diets or using pills, in contrast to women [37]. In addition, men are more responsive to treatment than women [38]. Nevertheless, social media exposure appears to exacerbate BI concerns and EDs more strongly in women than in men [39]. Other sociodemographic variables, such as age and educational level, have also been shown to be associated with BI concerns and ED symptoms. Mixed results have been found regarding the relationship between age and ED symptoms and BI concerns. For instance, BI concerns are associated with weight-related behaviours in both adolescents and emerging adults [40], whereas older people tend to report more positive body image over time [41]. Barakat et al. [13] reported that individuals with higher education followed stricter diets than those without higher education. However, Hay et al. [42] found that educational levels were similar between people with eating disorders and the general population.

Although earlier studies have isolated some of the single predictors associated with the development of EDs and BI concerns, the combined impact of these factors, as well as the differential contribution to each of these issues (EDs and BI concerns), has not been thoroughly explored. Previous research has found a relationship between BI concerns and EDs [13], but it is particularly important to identify the factors that contribute to the onset of each of these problems and to clarify the link between them. This study provides new insights into the common and specific protective and risk factors for both problem behaviours, ED symptoms and BI concerns, from a gender-differentiated perspective, filling an important gap in the previous literature. This study aims to examine the role of personal factors (perfectionism and life satisfaction), emotional factors (emotional regulation, anxiety, depression, and stress), and social factors (social media BI-related behaviours) in relation to ED symptoms and BI concerns, as well as to determine whether gender differences exist in the factors associated with these outcomes.

Based on a review of the literature, the following hypotheses were proposed. First, a relationship between BI concerns and ED symptoms is expected to be found, as previous studies have shown [13]. Second, it is hypothesised that women will exhibit greater ED symptomatology and BI concerns than men [34,35]. Third, age and educational level are expected to be associated with BI concerns and ED symptoms, with younger people and those with higher educational levels showing a greater risk for both problems [13,41]. Fourth, it is hypothesised that individuals who are more vulnerable to ED symptoms and BI concerns will have more emotional problems, greater perfectionism, more preoccupation with their BI on social media, and lower life satisfaction [12,14,23,33]. Fifth, the personal, emotional, and social factors involved in BI concerns and ED symptoms will have a different impact depending on gender. To our knowledge, no previous studies have analysed the common and specific predictors of ED symptoms and BI concerns by gender within the same investigation. However, consistent with Social Role Theory [43], it is expected that gender-specific pathways whereby perfectionism and life satisfaction deficits predict ED symptoms more strongly in women, whereas goal persistence is more salient in men.

## 2. Materials and Methods

### 2.1. Participants

The sample comprised 201 Spanish adults (47 men, 23.4%; and 154 women, 76.6%), ranging from 18 to 58 years (M = 28.26, SD = 8.34), with 88.6% aged between 18 and 40 years. Most participants had university-level education (64.4%). Among the remaining participants, 12.8% held a bachelor’s degree, 8.1% had completed higher education, 8.1% had intermediate education, 6.1% had completed basic education, and 0.5% had not finished basic education. There were 31.3% studying at the time of data collection. Baseline participant characteristics for the total sample, as well as for men and women separately, are shown in Table 1.

The inclusion criteria for the study were being over 18 years of age, belonging to a community sample, and having Spanish as their first language. The sole exclusion criterion was the lack of informed consent to participate in the study.

A priori power analysis was conducted using G*Power 3.1 to ensure an adequate sample size for multiple regression analyses. Assuming a medium effect size (f^2^ = 0.15), α = 0.05, and a power of 0.95, the required minimum sample was calculated to be 166 participants. Therefore, the final sample of 201 participants exceeded the minimum threshold, ensuring sufficient statistical power for the planned analyses.

### 2.2. Design

A cross-sectional design was used to analyse the relationship between personal, emotional, and social factors and ED symptoms and BI concerns, taking into account gender differences. The study was performed according to the Strengthening the Reporting of Observational Studies in Epidemiology (STROBE) guidelines [44].

### 2.3. Procedure

Participants were recruited through a convenience sample method. Data collection was conducted through Google Forms, which was distributed via WhatsApp and Instagram. The data was collected from January to February 2024. Approximately 640 people had access to the survey, of whom 201 responded, with a response rate of 31.41%. Participants were informed that the research aimed to identify the connection between BI and eating concerns, social media BI-related behaviours, as well as other personal and emotional factors that may account for concerns about BI and EDs. Additionally, participants consented online to participate in the research and were informed of their privacy rights and the completely voluntary nature of the study, with clarification that the data would only be used anonymously for research and publication purposes. Participation in the research was not remunerated. The study was conducted according to the guidelines of the Declaration of Helsinki. Ethical approval was obtained from the Animal Welfare and Research Ethics Committee of the University of La Laguna (CEIBA2020-0412; 4 July 2020).

The data was analysed using SPSS statistical software (version 25). Normal distribution of data was assessed using Kolmogorov–Smirnov (K-S) normality tests and descriptive statistics, kurtosis, and asymmetry. Although the K-S test indicated a non-normal distribution, the asymmetry (range −2 to +2) and kurtosis (range −5 to +5) indices were within the normality range [45]. Correlational analyses between sociodemographic variables, personal, emotional, and social factors, ED symptoms, and BI concerns were performed using Spearman’s correlation for categorical variables and Pearson’s correlation for continuous variables. Multiple regression analysis was used to identify the BI concerns factors that predicted ED symptoms. A multivariate analysis of variance (MANOVA) was conducted to examine gender differences in variables related to ED symptoms and BI concerns. A second MANOVA was performed to assess gender differences in social media body image-related behaviours. Finally, to identify the personal, emotional, and social variables that most significantly explained ED symptoms and BI concerns, multiple regression analyses were carried out. In the multiple regression analyses where the BI concerns factors were the criterion variables, gender was controlled for, while for ED symptoms, multiple regression analyses were conducted for men and women separately.

### 2.4. Instruments

Sociodemographic variables: The sociodemographic questionnaire collected information on participants’ gender, age, and educational level.

The instruments that assessed the primary outcome measures on ED symptoms and BI concerns were the following:

The Multidimensional Body–Self Relations Questionnaire (MBSRQ [46]; Spanish version [47]) has 45 items on a five-point Likert scale (1 = “strongly disagree” or “very dissatisfied”; 5 = “strongly agree” or “very satisfied”) that comprise evaluative, cognitive, and behavioural components of BI. The scale includes four factors: subjective importance of corporeality (SIC), fitness-oriented behaviours (FOB), self-assessed physical attractiveness (SPA), and care of external appearance (CEA). The overall scale has an internal consistency of 0.88, 94 for SIC, 81 for FOB, 71 for SPA, and 0.84 for CEA. In this study, the internal consistency was 0.63 for SIC, 0.93 for FOB, 0.83 for SPA, and 0.51 for CEA.

The Eating Attitudes Test (EAT-26 [48]; Spanish version [49]) consists of 26 items rated on a six-point Likert scale (1 = “never” to 6 = “always”). Responses were recoded as follows: 1–3 = 0; 4 = 1; 5 = 2; and 6 = 3. A total score of 19 or above indicates disordered eating attitudes and behaviours, distinguishing symptomatic from asymptomatic individuals. The original scale showed a reliability coefficient of α = 0.79; in the present study, internal consistency was α = 0.88.

The instruments that assessed the secondary outcome measures on personal, emotional, and social factors were the following:

The Eating Disorders Inventory (EDI-2 [50]; Spanish version [51]) consists of 91 items on a six-point Likert scale (0 = “never”; 5 = “always”), which evaluates attitudes and behaviours related to EDs. This study only uses the perfectionism scale. Internal consistency ranges from 0.83 to 0.93, and in this study, it was 0.85.

The Satisfaction with Life Scale (SWLS [52]; Spanish version [53]) has five items on a seven-point Likert scale (1 = “extremely dissatisfied”; 7 = “extremely satisfied”), which assesses an individual’s cognitive judgement of overall life satisfaction by comparing their life circumstances to a personal standard. The original scale has an internal consistency of 0.87 and a test–retest reliability of 0.82. In this study, the internal consistency was 0.86.

The Difficulties in Emotion Regulation Scale (DERS [54]; Spanish version [55]) has 36 items on a five-point Likert scale (1 = “almost never”; 5 = “almost always”) and assesses various aspects of emotional dysregulation, such as lack of emotional awareness (AWARENESS), impulse control difficulties (IMPULSE), non-acceptance of emotional responses (NON-ACCEPTANCE), goal-directed behaviour difficulties (GOALS), emotional clarity (CLARITY), and limited access to emotion regulation strategies (STRATEGIES). The internal consistency of the original global scale is 0.93. In this study, the internal consistency was 0.81 for awareness, 0.87 for impulse, 0.92 for non-acceptance, 0.87 for goals, 0.86 for clarity, and 0.85 for strategies.

The Depression, Anxiety, and Stress Scale (DASS-21 [56]; Spanish version [57]) comprises 21 items on a four-point Likert scale (0 = “did not apply to me”; 3 “applied to me very much or most of the time”), which evaluates negative emotional states through three subscales: depression, anxiety, and stress. The reliability of the scale is 0.87 for depression, 0.79 for anxiety, 0.83 for stress, and 0.93 for the total scale. In this study, the internal consistency was 85 for depression, 0.81 for anxiety, and 0.91 for stress.

Social media body image-related behaviours: In this study, four items were developed to assess social media BI-related behaviours. The behaviours assessed were refraining from posting content due to dissatisfaction with one’s BI (NON-POSTING), feeling insecure about one’s BI after viewing others’ content on social media (INSECURE), desire to have a body like other attractive individuals on social media (DESIRE), and editing or hiding disliked body parts when posting on social media (EDITING). Participants responded on a five-point Likert scale ranging from 1 “never” to 5 “always”.

## 3. Results

### 3.1. Preliminary Analyses

Table 2 shows the mean, standard deviation, and minimum and maximum scores for variables included in the study. The asymmetry index was around −2 and +2 and the kurtosis index was around −5 and +5 for all variables indicating the normal distribution of data. In general, the descriptive data indicated that the participants attributed considerable importance to the corporeality (SIC), although most of the respondents did not report EDs. In particular, of the total respondents, 18.4% (N = 37) of participants met the criteria for ED symptoms based on the cutoff point of the Eating Attitudes Test [48].

Correlational analyses between sociodemographic variables (age, gender, and educational level), ED symptoms, and the four factors of BI concerns (subjective importance of corporeality, fitness-oriented behaviours, self-assessed physical attractiveness, and care of external appearance) were performed. ED symptoms were only associated with gender (r = 0.17, *p* < 0.05), with women showing a positive association. No significant association was found between gender and BI concerns factors. However, some factors of BI concerns were related to age and educational level. Specifically, age was significantly correlated with subjective importance of corporeality (r = −0.17, *p* < 0.05), and the educational level correlated significantly with fitness-oriented behaviours (r = 0.25, *p* < 0.001).

Significant relationships between ED symptoms and BI concerns factors were found (Table 3). ED symptoms showed positive associations with subjective importance of corporeality (r = 0.52, *p* < 0.001), and care of external appearance (r = 0.43, *p* < 0.001), whereas negative associations with self-assessed physical attractiveness (r = −0.56, *p* < 0.001) and fitness-oriented behaviours (r = −0.16, *p* < 0.05) were found. An additional regression analysis was carried out to explore the link between BI concerns and ED symptoms in greater depth. The results showed that subjective importance of corporeality (β = 0.38, *p* < 0.001) and care of external appearance (β = 0.17, *p* < 0.01) predicted higher risk of EDs, while self-assessed physical attractiveness (β = −0.35, *p* < 0.001) and fitness-oriented behaviours (β = −0.18, *p* < 0.01) were negatively related to ED symptoms [F (4, 196) = 48.18, *p* < 0.001].

Significant positive relationships were also found between EDs and the four social media BI-related behaviours (Table 3); mainly, ED symptomatology was associated with not uploading photos to networks (r = 0.51, *p* < 0.001) and feeling insecure about one’s body (r = 0.48, *p* < 0.001). The four factors of BI concerns also showed significant relationships with social media BI-related behaviours. Self-assessed physical attractiveness was negatively associated with not uploading photos to networks (r = −0.56, *p* < 0.001), feeling insecure about one’s body (r = −0.56, *p* < 0.001) or desire to have a body like other attractive individuals (r = −0.36, *p* < 0.001).

In addition, ED symptoms were positively associated with difficulties in emotional regulation, mainly with non-acceptance of emotional responses (r = 0.54, *p* < 0.001), and limited access to emotion regulation strategies (r = 0.44, *p* < 0.001). A significant positive relationship of ED symptoms with anxiety (r = 0.43, *p* < 0.001) and perfectionism (r = 0.41, *p* < 0.001) was also found, while ED symptoms were negatively associated with life satisfaction (r = −0.41, *p* < 0.001). BI concerns factors showed positive and negative associations with personal and emotional factors. Subjective importance of corporeality was mainly related to increased perfectionism (r = 0.32, *p* < 0.001), goal-directed behaviours (r = 0.34, *p* < 0.001), and difficulties in accepting emotional responses (r = 0.30, *p* < 0.001). Care of external appearance was associated, above all, with difficulties in goal-directed behaviours (r = 0.25, *p* < 0.01) and greater stress (r = 0.22, *p* < 0.01). In contrast, self-assessed physical attractiveness and fitness-oriented behaviours were associated with higher life satisfaction (r = 0.56 and r = 0.42, *p* < 0.001, respectively) and lower difficulties in emotional regulation (Table 3).

### 3.2. Gender Differences

Given the correlation between gender and ED symptoms, a MANOVA was conducted to analyse gender differences in ED symptoms and the four factors related to BI concerns. The multivariate analysis found no significant differences [F (5, 195) = 1.24; *p* = 0.29]. However, univariate analyses revealed significant gender differences in EDs [F (1, 199) = 3.96, *p* < 0.05, η^2^ = 0.02, 1 − β = 0.51], with women scoring higher than men (Table 4). However, the effect size was small, according to Cohen’s *d*. No gender differences in BI concerns factors were confirmed.

On the other hand, a MANOVA was conducted to analyse gender differences in social media BI-related behaviours. The multivariate analysis revealed significant differences [F (4, 196) = 8.26; *p* < 0.001, η^2^ = 0.14, 1 − β = 0.99]. Univariate analyses indicated significant differences for not posting content due to BI dissatisfaction [F (1, 199) = 6.37; *p* < 0.05, η^2^ = 0.03, 1 − β = 0.71)], feeling insecure about BI when viewing others’ content on social media [F (1, 199) = 26.25; *p* < 0.001, η^2^ = 0.12, 1 − β = 0.99], the desire to have a body like attractive individuals on social media [F (1, 199) = 20.23; *p* < 0.001, η^2^ = 0.09, 1 − β = 0.99], and the editing or hiding disliked body parts when posting on social media [F (1, 199) = 9.78; *p* < 0.001, η^2^ = 0.05, 1 − β = 0.86], with women scoring higher on these measures than men (Table 5). These mean differences had a small effect size.

Additionally, multiple regression analyses were applied to identify predictors of each of the BI concerns factors. The regression models included personal and emotional factors and social media BI-related behaviours as predictors. The criterion variables were each of the BI concerns factors. Since there were no differences between men and women in BI concerns, the entire sample was included in the analyses. However, the gender variable was controlled since there were gender differences in all measures of social media BI-related behaviours (Table 6). The subjective importance of corporeality was explained by difficulty in persisting with goal-directed behaviours (β = 0.24, *p* < 0.001), a high level of perfectionism (β = 0.21, *p* < 0.01), and not posting images due to dissatisfaction with one’s body (β = 0.17, *p* < 0.05), accounting for 18.8% of the variance [F (3, 197) = 16.48, *p* < 0.001]. For fitness-oriented behaviours, life satisfaction (β = 0.38, *p* < 0.001) and high emotional awareness (β = −0.14, *p* < 0.05) explained 18.3% of the variance [F (2, 198) = 23.45, *p* < 0.001].

Regarding self-assessed physical attractiveness, the variables not posting content about oneself on social media due to dissatisfaction with BI (β = −0.37, *p* < 0.001), life satisfaction (β = 0.35, *p* < 0.001), feeling insecure about BI when viewing content from others on social media (β = −0.25, *p* < 0.01), lack of emotional awareness (β = −0.21, *p* < 0.001), and desire to have a body like other attractive individuals on social media (β = 0.15, *p* < 0.05) together explained 53.7% of the variance [F (5, 195) = 47.31, *p* < 0.001]. Finally, the variables difficulty in persisting with goal-directed behaviours (β = 0.17, *p* < 0.05) and not posting content about oneself on social media due to dissatisfaction with BI (β = 0.25, *p* < 0.001) accounted for 11.3% of the variance in care of external appearance [F (2, 198) = 13.70, *p* < 0.001].

Considering the gender differences in ED symptoms, multiple regression analyses were applied to identify which variables explained the ED symptoms for men and women separately (Table 7). The variables non-acceptance of emotional responses (β = 0.35, *p* < 0.001), not posting content about oneself on social media due to dissatisfaction with BI (β = 0.23, *p* < 0.001), a high level of perfectionism (β = 0.21, *p* < 0.001), and a low level of life satisfaction (β = −0.26, *p* < 0.001) together explained 46.2% of the variance in EDs among women [F (4, 149) = 33.79, *p* < 0.001]. In the case of men, difficulties in accepting emotional responses (β = 0.56, *p* < 0.001), and not posting images due to dissatisfaction with their body (β = 0.54, *p* < 0.001) also played an important role in ED symptoms; but interestingly, the ability to organise goal-directed behaviour (β = −0.21, *p* < 0.05) was associated with higher risk of EDs for men. These variables accounted for 72% of the variance of ED symptoms [F (3, 43) = 41.46, *p* < 0.001].

## 4. Discussion

This study aimed to examine the relationships between personal, emotional, and social factors and ED symptoms and BI concerns, by identifying the protective and risk factors most strongly associated with both issues from a gender-differentiated perspective. As expected, the first hypothesis was confirmed, suggesting significant associations between ED symptoms and BI concerns factors. People with ED symptomatology attributed more importance to the body and cared more about their appearance, while feeling less physically attractive and engaging in fitness behaviours to a lesser extent. This finding is particularly relevant, suggesting that the BI concerns can be an early indicator to consider in order to prevent more serious problems, such as EDs. Other studies have found that the negative evaluation of one’s appearance explains the relationship between eating attitudes and sociocultural attitudes toward appearance [58].

The results indicated that ED symptoms were associated with the fact of being a woman, but no gender differences were found in BI, which partially confirms the second hypothesis. Overall, data on the prevalence of EDs indicate that EDs are higher in women than in men, with an average lifetime prevalence of 8.4% for women and 2.2% for men [2]. Although in recent years studies have found no gender differences in EDs [38]. A tentative explanation could be that sociocultural factors related to new standards of beauty, fitness, and muscularity linked to health are exerting a minimising effect on gender differences. The third hypothesis was also partially confirmed; ED symptoms did not show any significant relationships with age and educational level, and only two factors of BI concerns were associated with these variables. Specifically, a weak association between placing more importance on corporeality and being younger was found. Previous studies have confirmed that young adults show less satisfaction with their BI than older adults [59]. Additionally, fitness-oriented behaviour was associated with higher education. In general, the university environment promotes physical exercise behaviours linked to body image and health [60]. These results could be explained by the social pressure that young people often feel to conform to beauty standards, which have increased in recent times with the growing use of social media [9] and it may be affecting younger people.

The fourth hypothesis, that people with more emotional problems, greater perfectionism, greater concern about their BI on social media, and lower life satisfaction are more vulnerable to ED symptoms and BI concerns, was confirmed. The ED symptoms and factors of subjective importance of corporeality and care of external appearance were mainly related to social media body image-related behaviours; so, people with ED symptoms were less likely to post photos, felt more insecure about their body image, and desired to have a body like other people on social media. Previous studies have found that consuming weight loss-related content on social media is associated with a worse negative BI and more EDs [7]. Individuals who considered themselves physically attractive were found to post more photos, felt less insecure, did not desire to have a body like those on social media, and did not edit their photos. This finding is particularly relevant given that the influence of physically attractive models on the BI of those who do not perceive themselves as attractive is confirmed, contributing to a further worsening of their body image and increasing their vulnerability to EDs. Additionally, difficulties in emotional regulation for ED symptoms and BI concerns were found. In particular, ED symptoms were associated with more difficulty accepting emotional responses, while the subjective importance of corporeality and care of external appearance were more closely linked to goal-direct behaviour difficulties. Previous research confirms these results; people with EDs show lower emotional acceptance, as well as impulsive control difficulties, goal-directed behaviour difficulties, and limited access to emotion regulation strategies [61]. On the contrary, a positive self-assessment of physical attractiveness was associated with a lower level of difficulty in accepting emotions and fitness-oriented behaviour was mainly related to a reduced difficulty in identifying emotions. Therefore, BI concerns have been shown to be associated with emotional dysregulation in people who perceive their image negatively, while people who are concerned with exercising and perceive themselves as physically attractive show better emotional regulation. Furthermore, both factors of body image (SPA and FOB) seem to promote life satisfaction, in line with previous studies [32].

A high level of perfectionism was also associated with the importance of corporeality. A tentative explanation for this finding could be that individuals who attribute excessive importance to corporeality, combined with high perfectionism, impose unrealistic appearance-related goals that are difficult to achieve, thus perpetuating dissatisfaction with their own body. Previous studies have found that these unrealistic standards, often promoted by the use of social media, have contributed in recent years to an increased prevalence of body dysmorphic disorder, mainly among young women [62]. Conversely, perceiving oneself as attractive and engaging in fitness-oriented behaviours was associated with increased emotional state awareness and higher life satisfaction. The ability to identify emotional states becomes an important predictor of the behaviours needed to maintain physical fitness. Previous research indicates that physical activity can improve both physical self-concept and life satisfaction [63]. In addition, although people who perceived themselves as attractive desired to have a body similar to that of other attractive people on social media, they did not feel insecure and did not refrain from posting content due to dissatisfaction with their bodies. Other studies have found that high self-esteem and body appreciation affect life satisfaction [64]. In the same way, a recent study reports that positive self-esteem or positive appraisal is associated with increased body satisfaction [65].

The findings also underlined commonality and gender differences in the variables involved in ED symptoms and BI concerns. Gender differences were identified in ED symptoms and social media BI-related behaviours, but not in BI concerns factors, partially confirming the fifth hypothesis. Women scored higher in ED symptoms and social media BI-related behaviours than men. Specifically, women desired to have an attractive body, felt insecure when comparing their physique to others in the media, and did not post content due to body dissatisfaction or edited their image to make themselves look more attractive. Other studies emphasise how prolonged exposure to idealised beauty standards on social media, which often promotes unrealistic BI, contributes to self-monitoring behaviours and increased body dissatisfaction, particularly among young women [66]. Exposure to objectifying images is associated with both body dissatisfaction and risky eating behaviours, encouraging image manipulation in photos posted on social networks, and this effect is stronger in young women [67,68]. Although recent studies have also found that internalisation of the male body ideal is associated with body dissatisfaction, emotional eating, desire for muscularity, and dietary restriction in adolescent boys [69].

Some personal, emotional, and social predictors involved in ED symptoms were common in men and women. Both men and women showed difficulties in accepting their emotions and did not usually post images on networks. Recent evidence links emotional regulation difficulties to body dissatisfaction in individuals at high risk for EDs [23]. Specifically, in a network meta-analysis study, non-acceptance of emotions and rumination were the factors most associated with EDs [22]. In addition to non-acceptance of emotions, other emotional regulation deficits, such as difficulties in impulse control, and limited access to emotional regulation strategies, have been linked to the likelihood of binge eating after an eight-month follow-up period [70]. In turn, not posting content on social media because of BI dissatisfaction may be explained by avoiding frequent comparisons with others that would exacerbate BI dissatisfaction and increase the drive for thinness, as proposed by [9]. This avoidance may reflect a protective strategy for individuals feeling vulnerable to judgement or dissatisfaction with their appearance.

However, gender differences were found in other variables explaining ED symptoms. Women at risk for ED symptoms were also characterised by high perfectionism and low life satisfaction, while men at risk for ED symptoms were characterised by no difficulties in engaging in goal-directed behaviours. Prior research has demonstrated that perfectionism is significantly related to disordered eating behaviours, often through the drive for thinness [16], which seems to be observed more commonly in women, probably due to social pressure to comply with certain standards of beauty. Sanzari et al. [7] corroborate these findings, indicating that perfectionism in women is positively associated with dieting behaviours, a pattern likely driven by both perfectionist tendencies and societal pressures surrounding body ideals. Other studies also find a negative relationship between EDs and life satisfaction [31], suggesting that women with ED symptoms may be dissatisfied with more areas of life than just their physical appearance.

Interestingly, in men, one predictor of ED risk was low difficulty in maintaining goal-directed behaviours. This finding may be explained by Social Role Theory [43], which suggests that men tend to be more agentic (task-oriented), whereas women are more communal (relationship-oriented). In this context, agentic orientation may have an adverse effect on men’s risk of developing EDs, as they tend to persist strongly in eating and body-related goals. The ideal male body is often associated with the development of muscularity and body toning, which requires rigorous exercise and strict dietary control to increase muscle mass and reduce body fat. The low difficulty in organising and sustaining goal-directed behaviours likely reflects this perseverance, reinforced by social norms valuing discipline and control in men [71]. However, such high persistence in reaching these goals can increase the risk of disordered eating, since the pressure to attain this ideal may foster excessive exercise behaviours and rigid and unhealthy patterns [72]. Recent studies indicate that men concerned with muscularity often show intense dedication to dietary control and bodybuilding, suggesting that the absence of difficulties in goal orientation does not necessarily imply well-being but may represent a risk factor within the male sociocultural context [73]. Therefore, low difficulty in goal-directed behaviour may reflect internalisation of gender-specific social demands, with important implications for ED prevention. In future studies, evaluating the reasons that lead men and women to restrict their food intake, along with including muscle-oriented scales to deepen understanding of factors underlying the risk of Eds, could be of interest.

As mentioned above, there were no gender differences in BI concerns. However, previous research has shown significant differences in body image based on gender. In a qualitative study that analysed the content and coping strategies regarding body image on social media among adolescents, it was found that boys avoided content that promoted beauty ideals and selected content that promoted their positive self-image, while girls not only did not select positive content but also were sceptical about such content [74]. Similarly, adult men rate an ideal body more positively when presented with their own face, but question thinness, hypermuscular bodies, and average-weight when presented with the bodies of others, while women rate ideal images of other people or images presented with their own face in a similar way [75]. Similarly, a recent study also found that adult females engaged in emotional eating as an avoidance strategy to a greater extent, which in turn resulted in a poorer body image compared to their male counterparts [76]. In our study, the most important personal, emotional, and social predictors of subjective importance of corporeality and care of appearance for men and women were difficulties in setting goals and avoiding posting images on social networks.

Results must be interpreted carefully due to study limitations. First, the study relied on a convenience sample that the researchers accessed through social media, resulting in certain sociodemographic characteristics, such as a young, female-dominant composition around 40 years of age and with higher education. These characteristics of the sample may obscure gender-specific, age-specific, and educational level-specific patterns or introduce significant variability that affects the interpretation of results related to BI concerns and ED symptoms, which limits the generalizability of findings. However, the findings of this study are in line with previous research showing a higher prevalence of EDs and BI in young, highly educated women [13,34]. Second, the instruments were administered online, and the response rate was low, which meant that it was not possible to access a more heterogeneous population, limiting the ability to extrapolate the results to other populations. In this study, one of the aims was to analyse behaviours related to body image in the media; hence, the participants were recruited online. However, the profile of people who respond to online surveys differs from that of the general population. Previous research had found lower response rates online than with pen and paper (6.7% vs. 86.5%), and a higher prevalence of EDs in students who responded online compared to those who responded on paper and pencil [77]. Further research is needed into the factors associated with BI concerns and ED symptoms through other assessment methods that capture the reality of these phenomena. Third, self-administered instruments, compared to other procedures such as interviews, may not capture the phenomenon under study and may produce biases due to a lack of understanding of the items or social desirability. For example, obese people have been found to report more binge eating through self-reporting than through interviews [78]. Fourth, some BI concerns scales, such as subjective importance of corporeality and care of external appearance, showed limited internal consistency, suggesting that it would be convenient to study the unidimensionality of these scales in a larger and more representative sample. Similarly, social media body image-related behaviours were assessed using four items, each of which measures a specific behaviour. Therefore, in the future, the design and validation of an instrument that captures body image behaviour on social media could be a relevant area of research. Finally, the study has focused on EDs as a continuous variable and could be of interest to analyse the impact of the factors studied in a clinical population. Future research should consider a wider sample size, including a more balanced sample in terms of age, gender, and educational level. Additionally, the inclusion of different types of EDs is essential to identify the type of BI concerns associated with different EDs and examine which dimensions prospectively predict changes in different EDs. Personal, emotional, and social factors could also have a differential impact depending on the different types of EDs.

Despite the limitations, the main strength of this study was that the common and specific risk and protective factors for ED symptoms and BI concerns were identified for both men and women. Furthermore, the assessment of body image-related behaviours on social media provided a comprehensive view of the phenomenon, highlighting the decisive role of social media in the risk of BI and EDs. Life satisfaction proved to be a protective factor associated with greater self-assessed physical attractiveness and fitness-oriented behaviours for men and women. In addition, perfectionism was a risk factor for EDs symptoms in women, while engaging in goal-directed behaviour was associated with EDs symptoms in men, indicating a paradoxical effect that needs to be studied in greater depth.

These findings have important practical implications. First, the results suggest that ED prevention programmes could be designed to focus on developing a more positive BI through psychoeducational programmes or literature on how social media influences EDs and BI. Second, intervention aimed at addressing BI concerns and ED symptoms could focus on promoting certain emotional regulation strategies, such as greater emotional awareness, consisting of the ability to identify and describe one’s own emotions and those of other people, which would promote a more positive self-image. Learning to set realistic goals would allow individuals to relativise the subjective importance of BI and external appearance. In addition, teaching people at risk of EDs to accept negative emotions, rather than using avoidant coping strategies focused on food restriction, would provide a way to address the problems of EDs. Previous studies highlight the importance of addressing patterns of behavioural avoidance and cognitive intensification of emotions in people with ED symptoms [79]. Third, the findings suggest the need to incorporate gender-specific components into intervention programmes, promoting greater life satisfaction and more realistic standards for women, as well as encouraging flexibility in goals for men.

## 5. Conclusions

A close relationship between ED symptoms and BI concerns factors was found. This study suggests that certain BI concerns factors, such as self-assessed physical attractiveness and fitness-oriented behaviours, may be protective of ED symptoms, while subjective importance of corporeality and care of external appearance were shown to be risk factors for ED symptoms. However, the personal, emotional, and social factors analysed had a stronger association with ED symptoms than with BI concerns. Particularly noteworthy was the close relationship found between ED symptoms and greater difficulties in emotional regulation, as well as avoiding behaviours on social networks such as posting photos, feeling insecure, or desiring to have a body like others. BI concerns also showed a significant link with social media body image-related behaviours. The predictive capacity of higher life satisfaction and emotional awareness on fitness-oriented behaviours and self-assessed physical attractiveness was particularly interesting. This insight highlights the importance of approaching body image from a more adaptive perspective focused on promoting health through exercise and fostering a more positive self-image.

Gender differences in ED symptoms remain important to highlight. Women continue showing a higher risk of EDs than men. Additionally, increased perfectionism and decreased life satisfaction were risk factors for women, while for men, a crucial risk factor was perseverance on goals. These emotional and behavioural tendencies suggest a complex interplay between ED symptoms and BI concerns in which social media plays a relevant role that needs to be explored in depth. Future research will identify other personal, emotional, and social factors involved in these issues in order to develop effective intervention programmes and promote the prevention of EDs and BI concerns at an earlier age.

## Figures and Tables

**Table 1 healthcare-13-01997-t001:** Baseline participants’ sociodemographic characteristics (N = 201).

	Total Participants (N = 201)	Women (N = 154)	Men (N = 47)
Gender (% women/men)	76.6/23.4		
Age (mean/SD)	28.26/8.34	28.33/8.48	28.04/7.92
Educational level (%)			
No formal education	0.5	0.6	_
Primary school	6.1	4.5	10.6
Intermediate vocational	8.1	8.4	6.4
Superior vocational	8.1	6.5	12.8
Secondary school ≥ 2 years	12.8	11.7	17.1
University ≥ 4 years	64.4	68.2	53.2

Note: SD standard deviation, % percentage.

**Table 2 healthcare-13-01997-t002:** Means, standard deviation, and minimum and maximum scores for ED symptoms, BI concerns factors, and social, personal, and emotional factors (N = 201).

	Mean	SD	Range (Min–Max)
ED symptoms	11.18	9.48	0–57
SIC	97.17	8.75	74–123
FOB	23.03	6.84	7–35
SPA	9.52	2.61	3–15
CEA	17.67	2.86	10–24
NON-POSTING	2.56	1.10	1–5
INSECURE	2.60	1.15	1–5
DESIRE	2.88	1.02	1–5
EDITING	2.29	1.19	1–5
PERFECTIONISM	15.52	6.78	1–30
LIFE SATISFACTION	23.54	5.80	8–35
DEPRESSION	3.23	3.55	0–19
ANXIETY	3.39	3.61	0–19
STRESS	5.86	4.64	0–19
AWARENESS	15.89	4.74	6–28
IMPULSE	12.12	4.55	6–29
NON-ACCEPTANCE	13.67	6.43	7–34
GOALS	13.29	4.82	5–25
CLARITY	10.13	3.63	5–22
STRATEGIES	14.36	4.98	7–29

Note: Eating disorder (ED) symptoms, subjective importance of corporeality (SIC), fitness-oriented behaviours (FOB), self-assessed physical attractiveness (SPA), care of external appearance (CEA), refraining from posting content due to dissatisfaction with one’s BI (NON-POSTING), feeling insecure about one’s BI after viewing others’ content on social media (INSECURE), desire to have a body like other attractive individuals on social media (DESIRE), editing or hiding disliked body parts when posting on social media (EDITING), lack of emotional awareness (AWARENESS), impulse control difficulties (IMPULSE), non-acceptance of emotional responses (NON-ACCEPTANCE), goal-directed behaviour difficulties (GOALS), emotional clarity (CLARITY), and limited access to emotion regulation strategies (STRATEGIES).

**Table 3 healthcare-13-01997-t003:** Correlations among ED symptoms, BI concerns factors, and social, personal, and emotional factors (N = 201).

	ED Symptoms	SIC	FOB	SPA	CEA
ED symptoms		0.52 ***	−0.16 *	−0.56 ***	0.43 ***
NON-POSTING	0.51 ***	0.31 ***	−0.10	−0.56 ***	0.31 ***
INSECURE	0.48 ***	0.31 ***	−0.18 *	−0.56 ***	0.30 ***
DESIRE	0.45 ***	0.31 ***	−0.07	−0.36 ***	0.29 ***
EDITING	0.36 ***	0.22 **	−0.23 ***	−0.38 ***	0.28 ***
PERFECTIONISM	0.41 ***	0.32 ***	−0.05	−0.21 **	0.20 **
LIFE SATISFACTION	−0.41 ***	−0.11	0.42 ***	0.56 ***	−0.05
DEPRESSION	0.36 ***	0.20 **	−0.07	−0.32 ***	0.10
ANXIETY	0.43 ***	0.15 *	−0.14 *	−0.25 ***	0.13
STRESS	0.35 ***	0.18 *	−0.11	−0.28 ***	0.22 ***
AWARENESS	0.19 **	−0.10	−0.24 ***	−0.31 ***	−0.11
IMPULSE	0.29 ***	0.12	−0.18 *	−0.28 ***	0.14
NON-ACCEPTANCE	0.54 ***	0.30 ***	−0.08	−0.34 ***	0.14
GOALS	0.41 ***	0.34 ***	−0.18 *	−0.31 ***	0.25 ***
CLARITY	0.28 ***	0.08	−0.19 **	−0.29 ***	0.11
STRATEGIES	0.44 ***	0.24 ***	−0.18 **	−0.30 ***	0.15 *

Note: Eating disorder (ED) symptoms, subjective importance of corporeality (SIC), fitness-oriented behaviours (FOB), self-assessed physical attractiveness (SPA), care of external appearance (CEA), refraining from posting content due to dissatisfaction with one’s BI (NON-POSTING), feeling insecure about one’s BI after viewing others’ content on social media (INSECURE), desire to have a body like other attractive individuals on social media (DESIRE), editing or hiding disliked body parts when posting on social media (EDITING), lack of emotional awareness (AWARENESS), impulse control difficulties (IMPULSE), non-acceptance of emotional responses (NON-ACCEPTANCE), goal-directed behaviour difficulties (GOALS), emotional clarity (CLARITY), and limited access to emotion regulation strategies (STRATEGIES). * *p* < 0.05; ** *p* < 0.01; *** *p* < 0.001.

**Table 4 healthcare-13-01997-t004:** Mean and standard deviation of ED symptoms and BI concern factors for men and women.

	Men (N = 47)		Women (N = 154)		F	η^2^	(1-Beta)
	Mean	SD	Mean	SD			
EDs	8.79	7.02	11.91	10.02	3.96 *	0.02	0.51
SIC	96.34	7.99	97.42	8.98	0.55	0.00	0.11
FOB	24.30	6.51	22.65	6.92	2.10	0.01	0.30
SPA	10.09	2.06	9.34	2.74	2.94	0.01	0.40
CEA	17.17	3.33	17.82	2.70	1.89	0.01	0.28

Note: Eating disorder (ED) symptoms, subjective importance of corporeality (SIC), fitness-oriented behaviours (FOB), self-assessed physical attractiveness (SPA), and care of external appearance (CEA). * *p* < 0.05.

**Table 5 healthcare-13-01997-t005:** Mean and standard deviation of social media BI-related behaviours for men and women.

	Men (N = 47)		Women (N = 154)		F	η^2^	(1-Beta)
	Mean	SD	Mean	SD			
NOT POSTING	2.21	1.08	2.67	1.08	6.37 *	0.03	0.71
INSECURE	1.89	1.05	2.82	1.09	26.25 ***	0.12	0.99
DESIRE	2.32	0.94	3.05	0.98	20.23 ***	0.09	0.99
EDITING	1.83	1.03	2.44	1	9.78 **	0.05	0.88

Note: Refraining from posting content due to dissatisfaction with one’s BI (NOT POSTING), feeling insecure about BI when viewing others’ content on social media (INSECURE), desire to have a body like other attractive individuals on social media (DESIRE), and editing or hiding disliked body parts when posting on social media (EDITING). * *p* < 0.05; ** *p* < 0.01; *** *p* < 0.001.

**Table 6 healthcare-13-01997-t006:** Multiple regression analyses of social, personal, and emotional factors on BI concerns factors controlling gender.

	β	95% CI for B Lower/Upper	R^2^	F	VIF
DV: Subjective importance of corporeality			0.19	16.48 ***	
NOT POSTING	0.17 *	0.26/2.46			1.21
GOALS	0.24 ***	0.19/0.68			1.14
PERFECTIONISM	0.21 **	0.09/0.44			1.15
DV: Fitness-oriented behaviours			0.18	23.45 ***	
AWARENESS	−0.14 **	−0.39/−0.01			1.07
LIFE SATISFACTION	0.38 ***	0.30/0.60			1.07
DV: Self-assessed physical attractiveness			0.54	47.31 ***	
NOT POSTING	−0.37 ***	−1.21/−0.54			2.24
INSECURE	−0.25 **	−0.92/−0.20			2.85
DESIRE	0.15 *	0.04/0.72			1.93
AWARENESS	−0.21 ***	−0.17/−0.06			1.11
LIFE SATISFACTION	0.35 ***	0.11/0.20			1.31
DV: Care of external appearance			0.11	13.70 ***	
NOT POSTING	0.25 ***	0.29/1.02			1.11
GOALS	0.17 *	0.02/0.19			1.11

Note: DV = dependent variable, CI = confidence interval, VIF = variance inflation factor, refraining from posting content due to dissatisfaction with one’s body image (NOT POSTING), goal-directed behaviour difficulties (GOALS), feeling insecure about body image when viewing others’ content on social media (INSECURE), lack of emotional awareness (AWARENESS), and desire to have a body like other attractive individuals on social media (DESIRE). * *p* < 0.05; ** *p* < 0.01; *** *p* < 0.001.

**Table 7 healthcare-13-01997-t007:** Multiple regression analyses of social, personal, and emotional factors on ED symptoms for women and men.

	β	95% CI for B Lower/Upper	R^2^	F	VIF
DV: ED symptoms for women			0.46	33.79 ***	
NON-ACCEPTANCE	0.35 ***	0.37/0.78			1.15
NOT POSTING	0.23 ***	0.97/3.33			1.19
LIFE SATISFACTION	−0.26 **	−0.67/−0.24			1.11
PERFECTIONISM	0.21 ***	0.13/0.48			1.09
DV: ED symptoms for men			0.72	41.46 ***	
NON-ACCEPTANCE	0.56 **	0.33/0.71			1.73
NOT POSTING	0.54 ***	2.32/4.70			1.38
GOALS	−0.21 *	−0.68/−0.03			1.49

Note: DV = dependent variable, CI = confidence interval, VIF = variance inflation factor, non-acceptance of emotional responses (NON-ACCEPTANCE), refraining from posting content due to dissatisfaction with one’s body image (NOT POSTING), and goal-directed behaviour difficulties (GOALS). * *p* < 0.05; ** *p* < 0.01; *** *p* < 0.001.

## Data Availability

The data presented in this study are available on request from the corresponding author due to privacy issues. However, the raw data have been annexed to the journal for internal evaluation.

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
