# Peer review of "Exploring the Involvement of Personal and Emotional Factors and Social Media Body Image-Related Behaviours on Eating Disorder Symptoms and Body Image Concerns in Women and Men"

_healthcare, 2025, doi:10.3390/healthcare13161997_

Round 1
Reviewer 1 Report
Comments and Suggestions for Authors
Abstract
According to MDPI journal writing guidelines, the abstract should not exceed 200 words. The study's abstract is too long.
The method section in the abstract is insufficient; only information about the groups participating in the study is provided. The applications should also be described.
Introduction
The introduction adequately describes the study.
The originality of the research is not stated.
A clear research question or hypothesis on which the research is based is not defined.
After discussing the effects of social media on body image, the reason for the topic is not explained.
The originality of the research and its hypotheses should also be included in the introduction.
Method
The research model was not defined.
No normality test was not disclosed.
Whether the number of participants was representative of the population was not discussed.
Power analysis was not conducted.
Which university should be written instead of "our university" in 154 lines?
Inclusion and exclusion criteria were not specified.
Insufficient internal consistency, external validity, and reliability information for the scales used.
Results
In Table 1, some are in uppercase, some in lowercase. Which is correct? (The same is in Table 2.)
Lines 254-256 of the findings should contain only findings and no comments.
Lines 270-278 and 279-281 should contain no comments
Discussion
Because the study's hypothesis was not clearly stated, it was not interpreted whether the findings supported the hypothesis.
Comparisons with the literature are insufficient. Only similar studies are cited, and cause-and-effect relationships are not explained in detail.
The study's strengths and limitations are too brief and not adequately explained. There are no strengths off study.
References
30 references, some using commas and others using the ampersand. References should be written according to the spelling guide.
Author Response
Abstract
According to MDPI journal writing guidelines, the abstract should not exceed 200 words. The study's abstract is too long.
Please, excuse the length of the abstract. The information has been summarised in relation to the previous abstract, although with the suggestion to include more details of the method, we have not been able to reduce it to less than 233 words.
The method section in the abstract is insufficient; only information about the groups participating in the study is provided. The applications should also be described.
Thank you for the suggestion. The objective has been clarified and the study design has been included.
Introduction
The introduction adequately describes the study.
Thanks to the reviewer for appreciating the theoretical review of these topics.
The originality of the research is not stated.
Thank you for the suggestion. In the last paragraph of the introduction, we have highlighted the relevance of the study. Although earlier studies have isolated some of the single predictors included in our study, the combined impact of the various predictors has not been analysed. This study provides new insights into the common and specific risk factors for both problem behaviours, EDs symptoms and BI concerns, from a gender-differentiated perspective, filling an important gap in the previous literature.
A clear research question or hypothesis on which the research is based is not defined.
The reviewer is right. The objective and hypothesis have been clarified in the last paragraph of the introduction.
After discussing the effects of social media on body image, the reason for the topic is not explained.
The importance of examining the role of social media in body image and eating disorders has been clarified. Thank you for the suggestion.
The originality of the research and its hypotheses should also be included in the introduction.
As suggested by the reviewer, these issues have been incorporated into the introduction.
Method
The research model was not defined.
A section on design has been included to clarify the cross-sectional nature of the research.
No normality test was not disclosed.
We apologise for not providing this important information. In the second paragraph of the procedure section, the normality analyses used have been included. At the end of this report, we have presented additional information on kurtosis and asymmetry indices for the different variables to clarify the question asked by the reviewer. Additionally, we have included gender-based differential analyses using the Mann-Whitney U test, which clearly shows that the results follow the same trend.
Whether the number of participants was representative of the population was not discussed.
The reviewer is right. In the procedure section, it has been clarified that the participants were a convenience sample.
Power analysis was not conducted.
We apologise for omitting this information. It has now been included at the end of the participants’ section.
Which university should be written instead of "our university" in 154 lines?
We did not include the name of the university for the blind peer review process. However, it has now been included as suggested by the reviewer.
Inclusion and exclusion criteria were not specified.
Thank you for the recommendation. The inclusion and exclusion criteria have been incorporated into the participants’ section.
Insufficient internal consistency, external validity, and reliability information for the scales used.
The scales used have demonstrated adequate psychometric criteria in previous studies and have been widely used in the literature. Indeed, in our study, there are only two scales, subjective importance of corporeality and care of external appearance, which have shown internal consistency of .63 and .51, respectively. The remaining scales all exceed .80 internal consistencies. This point has been discussed in the limitations section.
Results
In Table 1, some are in uppercase, some in lowercase. Which is correct? (The same is in Table 2.)
The abbreviations for scales with longer names appeared in uppercase, while scales with their original names appeared in lowercase letters. However, we have corrected the names of all scales so that they appear in uppercase in all tables.
Lines 254-256 of the findings should contain only findings and no comments.
The comments on the results have been removed, as suggested by the reviewer.
Lines 270-278 and 279-281 should contain no comments
The comments on the results have been removed, as suggested by the reviewer.
Discussion
Because the study's hypothesis was not clearly stated, it was not interpreted whether the findings supported the hypothesis.
Thank you for the suggestion, which helps to clarify the study. The hypotheses have been included at the end of the introductory section, and the results have been debated in the discussion section according to whether they confirm or refute the hypotheses.
Comparisons with the literature are insufficient. Only similar studies are cited, and cause-and-effect relationships are not explained in detail.
Although we have not established cause-and-effect relationships due to the cross-sectional nature of the study, we have included a more in-depth discussion of the results found in the literature, which in some cases contradict the findings of our research.
The study's strengths and limitations are too brief and not adequately explained. There are no strengths off study.
According to the reviewer's suggestion, the limitations and strengths of the study have been revised.
References
30 references, some using commas and others using the ampersand. References should be written according to the spelling guide.
Thanks to the reviewer for detecting the mistakes in the references. We have made the appropriate corrections.
Reviewer 2 Report
Comments and Suggestions for Authors
Title: Exploring the involvement of personal and emotional factors and social-media body-image-related behaviours on eating-disorder symptoms and body-image concerns
The study tackles an important, timely topic—the interplay between psychological factors, social-media use, and eating-disorder (ED) risk. The sample is adequate (N = 201) and the statistical approach is generally appropriate. Nonetheless, the manuscript requires substantive revisions to meet reporting and methodological standards and to sharpen its theoretical contribution.
1 Abstract
1.1 Abbreviations such as “EDs” and “BI” appear without full terms at first mention. Please spell out each abbreviation when it first occurs and remove any unused acronyms (e.g., “SM” for social media, which is never reused).
1.2 Two adjacent sentences describe social-media influences in almost identical wording. Condense to one clear statement, e.g., “Social-networking sites amplify idealised body-image promotion, exacerbating body dissatisfaction and disordered eating.”
2 Introduction
2.1 The manuscript cites Social Role Theory but does not connect it to the gender hypotheses. Add one or two linking sentences—e.g., “Consistent with Social Role Theory (Wood & Eagly, 2012), we expected gender-specific pathways whereby perfectionism and life-satisfaction deficits would predict ED symptoms more strongly in women, whereas goal-persistence would be more salient in men.”
2.2 Clarify the unique gap addressed: earlier studies have isolated single predictors, whereas this study explores their combined impact.
3 Methods – Participants
3.1 Recruitment details absent. State explicitly that participants were recruited via online convenience sampling and that survey links were distributed through WhatsApp, Telegram, Instagram and Twitter.
3.2 Sample-size justification. Provide an a-priori power analysis (e.g., G*Power 3.1, f² = 0.15, α = 0.05, power = 0.95) to demonstrate that N = 201 exceeds the required sample of 166.
3.3 Procedure:Report the total number of invitations, the number of completed surveys, and the resulting response rate (e.g., “230 distributed, 201 usable; response rate = 87.4%”). Describe exclusion criteria for incomplete data.
3.4 Instruments:Move the description of the custom four-item Social-Media Body-Image Behaviour questionnaire into the Instruments subsection for coherence. Add copyright / permission statements for proprietary measures (EAT-26, DASS-21, EDI-2, etc.).
4 Results
4.1 Interpret effect sizes. For instance, Table 3 reports η² = 0.02; note that this is a small effect under Cohen’s conventions.
4.2 Reformat Tables 3 and 5: align variable labels and statistics, keep decimal precision consistent, and label columns clearly.
5 Discussion
5.1 The link between men’s “goal-directed behaviours” (GOALS) and ED risk is over-interpreted. Acknowledge that this variable may index drive for muscularity rather than thinness and recommend inclusion of muscle-oriented scales in future studies.
5.2 Expand the limitations paragraph: emphasise convenience sampling, young female-dominant composition, and sole reliance on self-report instruments.
6 Conclusions:Re-state practical implications more concretely: prevention programmes should combine emotion-acceptance training with social-media literacy, and should incorporate gender-tailored components (perfectionism/life satisfaction for women; goal flexibility for men).
Author Response
The study tackles an important, timely topic—the interplay between psychological factors, social-media use, and eating-disorder (ED) risk. The sample is adequate (N = 201) and the statistical approach is generally appropriate. Nonetheless, the manuscript requires substantive revisions to meet reporting and methodological standards and to sharpen its theoretical contribution.
Thank you for your appreciation of our work and your suggestions.
1 Abstract
1.1 Abbreviations such as “EDs” and “BI” appear without full terms at first mention. Please spell out each abbreviation when it first occurs and remove any unused acronyms (e.g., “SM” for social media, which is never reused).
We thank the reviewer for the suggestion. The abbreviations have been incorporated at the first mention of terms.
1.2 Two adjacent sentences describe social-media influences in almost identical wording. Condense to one clear statement, e.g., “Social-networking sites amplify idealised body-image promotion, exacerbating body dissatisfaction and disordered eating.”
Thank you for your contribution. We have proceeded to simplify the repetitive expressions.
2 Introduction
2.1 The manuscript cites Social Role Theory but does not connect it to the gender hypotheses. Add one or two linking sentences—e.g., “Consistent with Social Role Theory (Wood & Eagly, 2012), we expected gender-specific pathways whereby perfectionism and life-satisfaction deficits would predict ED symptoms more strongly in women, whereas goal-persistence would be more salient in men.”
We appreciate your contribution. It certainly improves our hypothesis about gender differences.
2.2 Clarify the unique gap addressed: earlier studies have isolated single predictors, whereas this study explores their combined impact.
Thank you for your suggestion, which will certainly help to clarify the contribution to research in this field. In the last paragraph of the introduction, we have included this information.
3 Methods – Participants
3.1 Recruitment details absent. State explicitly that participants were recruited via online convenience sampling and that survey links were distributed through WhatsApp, Telegram, Instagram and Twitter.
The reviewer is right, thank you for your appreciation. We have clarified that the sample was a convenience sample.
3.2 Sample-size justification. Provide an a-priori power analysis (e.g., G*Power 3.1, f² = 0.15, α = 0.05, power = 0.95) to demonstrate that N = 201 exceeds the required sample of 166.
We apologise for omitting this information. It has now been included at the end of the participants’ section.
3.3 Procedure:Report the total number of invitations, the number of completed surveys, and the resulting response rate (e.g., “230 distributed, 201 usable; response rate = 87.4%”).
In accordance with the reviewer's observation, we have included the number of invitations and the response rate.
Describe exclusion criteria for incomplete data.
Thank you for the suggestion. The inclusion and exclusion criteria have been included in the participants’ section.
3.4 Instruments:Move the description of the custom four-item Social-Media Body-Image Behaviour questionnaire into the Instruments subsection for coherence. Add copyright / permission statements for proprietary measures
The description of the four items on social media body image-related behaviours has been included in the instruments section.
The remaining instruments used in the study have been published in scientific journals, which allows the scales to be used without copyright restrictions.
4 Results
4.1 Interpret effect sizes. For instance, Table 3 reports η² = 0.02; note that this is a small effect under Cohen’s conventions.
Thank you for your contribution. It has been clarified that the effect was small in the different contrasts.
4.2 Reformat Tables 3 and 5: align variable labels and statistics, keep decimal precision consistent, and label columns clearly.
Thank you for your suggestion. The tables have been checked to maintain the same format in each one.
5 Discussion
5.1 The link between men’s “goal-directed behaviours” (GOALS) and ED risk is over-interpreted. Acknowledge that this variable may index drive for muscularity rather than thinness and recommend inclusion of muscle-oriented scales in future studies.
This relationship between goal-directed behaviours and EDs in men has been discussed in greater detail, as suggested by the reviewer.
5.2 Expand the limitations paragraph: emphasise convenience sampling, young female-dominant composition, and sole reliance on self-report instruments.
The limitations of the study have been discussed at greater depth.
6 Conclusions:Re-state practical implications more concretely: prevention programmes should combine emotion-acceptance training with social-media literacy, and should incorporate gender-tailored components (perfectionism/life satisfaction for women; goal flexibility for men.
Thanks to the reviewer for the relevant suggestions provided for the conclusion section.
Reviewer 3 Report
Comments and Suggestions for Authors
Major comments:
- Introduction:
a. Overall, I found the introduction to be clear enough.
b. Lines 100-104: I do not think it fits with the main topic discussed, as the authors only discussed ‘perfectionism’ from the EAT-26 questionnaire and did not discuss eating behaviour further. In addition, it is not coherent with the following lines.
2. Methods:
a. Add the inclusion and exclusion criteria for 2.1.
b. Please provide the sample calculation
c. Please include the reporting guideline used, e.g. STROBE
d. The authors used a ‘custom questionnaire’. Please provide scientific evidence to support the questionnaire's validity and reliability.
e. Are there any specific reasons for using only perfectionism in the EAT-26? It is an unusual practice to include only one subscale.
f. Instruments: I suggest differentiating between the primary outcomes and the dependent variable
3. Results:
a. Although the authors have explained the step-by-step analysis, overall, I found the results hard to follow. I would suggest simplifying the narratives by mentioning only important points/highlights. The insignificant results could be omitted from the narration.
b. Please include the baseline characteristics table.
c. Tables:
- Table 1. I would suggest including only the mean SD for normal data or the median (IQR) for non-normal data. Range is not needed.
- For the long tables, I suggest following the examples in this article DOI: 10.1016/j.eclinm.2024.102645
- To make it easier to read, I suggest using only (*) to indicate significance, without the need to differentiate between *,** and ***. However, the decision rests with the authors.
d. I have a concern regarding a large portion of the results and discussion being focused on gender differences. This was not the focus of this study, as mentioned in the aims. I suggest providing a proportional result and discussion in line with the aims. If the authors plan to discuss gender differences as a significant proportion of the paper, the authors might want to consider changing the title and aims.
- Discussion:
a. Similar to comments from the results, I found the focus of this discussion mainly on gender differences. I suggest rewriting the conclusion and utilising headings, focusing on the title, e.g. 4.1. Involvement of personal and emotional factors, 4.2. Involvement of social media body image-related behaviours.
b. Conclusion on this is questionable, “This study suggests that certain BI concerns factors, such as self-assessed physical attractiveness and fitness oriented behaviours, may be protective of EDs symptoms”: If I read it correctly, I think the authors mentioned this conclusion based on a bivariate analysis, and did not explore further in the multivariate analyses and discussion.
Minor comments:
- Abstract: define the study, e.g. cross-sectional
- Use consistent abbreviations, e.g. eating disorders (EDs); some were abbreviated, others were not.
- Line 87: Did the authors mean ‘on one side’?
- Line 245: Did the authors mean ‘on the other hand’?
- Participants (2.1) belong to the result section
Author Response
Introduction:
- Overall, I found the introduction to be clear enough.
Thanks to the reviewer for appreciating the theoretical review of these topics.
- Lines 100-104: I do not think it fits with the main topic discussed, as the authors only discussed ‘perfectionism’ from the EAT-26 questionnaire and did not discuss eating behaviour further. In addition, it is not coherent with the following lines.
Thank you for your comment. The aim of those lines was to highlight how an adequate and balanced diet leads to greater life satisfaction compared to an altered intake, usually due to excess or deficiency, which is characteristic of eating disorders. This point has been clarified.
- Methods:
- Add the inclusion and exclusion criteria for 2.1.
The reviewer is right. The inclusion and exclusion criteria have been included in the participants’ section.
- Please provide the sample calculation
The power analysis has now been included at the end of the participants’ section.
- Please include the reporting guideline used, e.g. STROBE
A reference to the STROBE guideline has been included in the design section.
- The authors used a ‘custom questionnaire’. Please provide scientific evidence to support the questionnaire's validity and reliability.
The wording of the instrument that assessed social media body image-related behaviours has been corrected. It has been clarified that only four types of behaviours were recorded using a single item. The limitations section emphasises the importance of developing specific tools to assess these behaviours.
- Are there any specific reasons for using only perfectionism in the EAT-26? It is an unusual practice to include only one subscale.
The EAT-26 was administered in its entirety to assess disordered eating attitudes and behaviours and to distinguish between symptomatic and asymptomatic individuals.
The EDI-2 provides relevant information on symptoms related to anorexia and bulimia nervosa but we had already assessed these symptoms using the EAT-26. The EDI-2 questionnaire is a long questionnaire with 91 items that measures eleven scales, including obsession with thinness, body dissatisfaction, fear of maturity, and perfectionism, among others. In this study, we were only interested in evaluating personal factors using a short scale, so only the perfectionism items were selected. The instrument has appropriate psychometric criteria for each of the scales, which means that a single scale can be used without compromising its construct validity. Perfectionism trait has been shown to be a transdiagnostic factor in the development of BI and EDs.The relevance of this trait has been justified in the introduction section.
- Instruments: I suggest differentiating between the primary outcomes and the dependent variable
Thank you for the suggestion. The description of the instruments has been organised according to primary and secondary outcomes.
- Results:
- Although the authors have explained the step-by-step analysis, overall, I found the results hard to follow. I would suggest simplifying the narratives by mentioning only important points/highlights. The insignificant results could be omitted from the narration.
Thank you for the suggestion. We have removed the narrative of the results and summarised the most significant findings.
- Please include the baseline characteristics table.
Baseline participants’ characteristics for total sample as well as separated men and women are included in Table 1.
- Tables:
- Table 1. I would suggest including only the mean SD for normal data or the median (IQR) for non-normal data. Range is not needed.
We thank the reviewer for their comment. The description of the normality of the measurements has been included. The asymmetry index was around -2 and +2 and the kurtosis index was around -5 and +5 for all variables indicating the normal distribution of data. However, we have retained the range of the variables in order to provide a brief overview of the dispersion or variability of the data.
- For the long tables, I suggest following the examples in this article DOI: 10.1016/j.eclinm.2024.102645
The tables are presented in a similar way to that proposed by the reviewer in the article. In Table 6 and Table 7, which presents the multiple regression analyses, we have made some adjustments to clarify the presentation of the results.
- To make it easier to read, I suggest using only (*) to indicate significance, without the need to differentiate between *,** and ***. However, the decision rests with the authors.
Although the reviewer is right in terms of greater visibility of the numerical data if we remove some of the asterisks, we consider the exact level of significance to be relevant.
- I have a concern regarding a large portion of the results and discussion being focused on gender differences. This was not the focus of this study, as mentioned in the aims. I suggest providing a proportional result and discussion in line with the aims. If the authors plan to discuss gender differences as a significant proportion of the paper, the authors might want to consider changing the title and aims.
The reviewer is right. We have included the gender issue in the title and aims.
The discussion focused on the results obtained from the overall sample, followed by a debate on the results based on gender.
Discussion:
- Similar to comments from the results, I found the focus of this discussion mainly on gender differences. I suggest rewriting the conclusion and utilising headings, focusing on the title, e.g. 4.1. Involvement of personal and emotional factors, 4.2. Involvement of social media body image-related behaviours.
The discussion section has been reorganised according to the hypotheses included at the end of the introduction. We hope that this will improve the flow of the description and discussion of the main findings and their theoretical and applied implications.
- Conclusion on this is questionable, “This study suggests that certain BI concerns factors, such as self-assessed physical attractiveness and fitness oriented behaviours, may be protective of EDs symptoms”: If I read it correctly, I think the authors mentioned this conclusion based on a bivariate analysis, and did not explore further in the multivariate analyses and discussion.
The reviewer is right, the relationship between BI concerns and EDS was analysed using bivariate correlations. Based on your suggestion, we have included an additional regression analysis exploring BI concerns as predictors of EDS.
Minor comments:
Abstract: define the study, e.g. cross-sectional
The nature of the study has been included
Use consistent abbreviations, e.g. eating disorders (EDs); some were abbreviated, others were not.
The amendment has been made.
Line 87: Did the authors mean ‘on one side’?
This expression has been removed.
Line 245: Did the authors mean ‘on the other hand’?
This expression has been removed.
Participants (2.1) belong to the result section
The reviewer is right in that the description of the participants could be included in the results section. However, the journal template places the participants section within Materials and Methods, so we have decided to keep it in its current position.
Round 2
Reviewer 1 Report
Comments and Suggestions for Authors
Thank you for your revisions.
Author Response
Thank you for considering the publication of our manuscript.
Reviewer 2 Report
Comments and Suggestions for Authors
The authors have diligently addressed all points raised during the initial review. The revisions—particularly the expanded discussion of gender differences in goal-directed behaviors, clarification of methodological limitations, and enhanced contextualization of social media's role—have significantly strengthened the manuscript. Statistical analyses remain robust, and the conclusions are now better aligned with the data. The manuscript now meets the journal’s standards for novelty, clarity, and contribution to understanding gender-specific risk factors for eating disorders and body image concerns. It is suggested that this manuscript be accepted after carefully checking the format of references and grammar.
Author Response
Thank you for considering the publication of our manuscript. We have revised the format of the references, the numerical alignment of the tables, and the grammar (marked in red in the text).
Please do not hesitate to contact us if you have any further questions.
Best regards,
Reviewer 3 Report
Comments and Suggestions for Authors
All comments have been addressed accordingly.
Author Response
Thank you for considering the publication of our manuscript.
Best regards,